# Emergent constraints on carbon budgets as a function of global warming

Peter M. Cox [1,2] ✉, Mark S. Williamson [1,2], Pierre Friedlingstein [1,2,3], Chris D. Jones [4], Nina Raoult[1,2], Joeri Rogelj [5,6] & Rebecca M. Varney [1,2]

Earth System Models (ESMs) continue to diagnose a wide range of carbon budgets for each level of global warming. Here, we present emergent constraints on the carbon budget as a function of global warming, which combine the available ESM historical simulations and future projections for a range of scenarios, with observational estimates of global warming and anthropogenic $CO_2$ emissions to the present day. We estimate mean and likely ranges for cumulative carbon budgets for the Paris targets of 1.5 °C and 2 °C of global warming of 812 [691, 933] PgC and 1048 [881, 1216] PgC, which are more than 10% larger than the ensemble mean values from the CMIP6 models. The linearity between cumulative emissions and global warming is found to be maintained at least until 4 °C, and is consistent with an effective Transient Climate Response to Emissions (eTCRE) of 2.1 [1.8, 2.6] °C/1000PgC, from a global warming of 1.2 °C onwards.

The discovery of a near-proportionality between cumulative anthropogenic carbon dioxide emissions and global warming since pre-industrial times[1-3] is arguably the most important policy-relevant simplification of climate change science in the last 25 years. It has allowed climate change policy to be framed in terms of the total carbon budgets consistent with stabilising global warming at a given level, without the complications associated with the timing of emission cuts[4]. The carbon budgets associated with levels of global warming which are consistent with the Paris Agreement (1.5 °C and 2 °C), are now driving global climate policy and also national Net Zero pledges[5]. Reliable estimation of the remaining carbon budgets for these levels of global warming, has therefore become of critical importance for climate change policy[6].

However, the carbon budgets consistent with the Paris Targets remain uncertain. The latest CMIP6 Earth System Models (ESMs) continue to diagnose a wide range of carbon budgets for each level of global warming, in large part due to differences in the climate sensitivity to $CO_2$ across the models[7]. The recent Intergovernmental Panel on Climate Change 6th Assessment Report (IPCC AR6) attempted to

derive more useful ranges for the remaining carbon budgets[8] by mixing observational constraints on transient climate sensitivity[9], with ESM ranges for the sensitivity of land and ocean carbon sinks to climate and $CO_2$[10]. The IPCC AR6 estimates remaining carbon budgets from the beginning of 2020 of 140 PgC and 370 PgC for 1.5 °C and 2 °C respectively. Given that global (fossil fuel plus land-use) emissions are over 10 PgC per year[11], this implies little more than a decade of anthropogenic $CO_2$ emissions at current levels to avoid passing 1.5 °C, and therefore casts serious doubt on the achievability of the Paris Agreement without a major temperature overshoot[12].

In this paper we present an alternative approach which combines the available CMIP6 ESM historical and future projections[13], and observational estimates of global warming and anthropogenic $CO_2$ to the present day[11], to derive emergent constraints on carbon budgets as a function of global warming. Emergent constraints rely on a relationship which is evident across an ensemble of models, between aspects of future climate projections and simulated variations in the contemporary and past climate[14,15]. Published emergent constraints predominantly make use of observed variations in time, such as

[1]Faculty of Environment, Science, and Economy, University of Exeter, Exeter, UK. [2]Global Systems Institute, University of Exeter, Exeter, UK. [3]Laboratoire de Météorologie Dynamique/Institut Pierre-Simon Laplace, CNRS, Ecole Normale Supérieure/Université PSL, Sorbonne Université, Ecole Polytechnique, Paris, 75231, France. [4]Met Office-Hadley Centre, Fitzroy Road, Exeter, UK. [5]Centre for Environmental Policy and Grantham Institute - Climate Change and Environment, Imperial College London, London, UK. [6]Energy, Climate and Environment Program, International Institute for Applied Systems Analysis, Laxenburg, Austria. ✉e-mail: p.m.cox@exeter.ac.uk

seasonal cycles[16,17], interannual variability[18,19], and longer-term trends[20-22]. To constrain future carbon budgets, we instead use an observational constraint on the Specific Carbon Budget to the current day, i.e. the cumulative carbon emissions in PgC per °C of global warming[23].

We do not presume a linear relationship between emissions and global warming. Instead, we look for an emergent relationship across models between the cumulative carbon emissions per degree of global warming up to the current day, and the specific carbon budgets for different future levels of global warming. As we are interested in constraining the global carbon budgets for the Paris climate targets of 1.5 °C and 2 °C, we deliberately span scenarios with different future trajectories of non-$CO_2$ forcing factors. As a result, our specific carbon budgets are not the inverse of the $CO_2$-only 'Transient Climate Response to Emissions' (TCRE), but are instead the inverse of the effective TCRE (eTCRE), which includes the effects of non-$CO_2$ forcing factors. The latter is much more useful for UNFCCC climate policy, which needs to account for uncertainties in future atmospheric concentrations of non-$CO_2$ greenhouse gases and aerosols, as well as $CO_2$.

The CMIP6 ESM projections that we consider are driven by Shared Socioeconomic Pathway (SSP) concentration scenarios[24]. This approach was adopted in CMIP5 and CMIP6 as it allows ESMs to use comparable concentration scenarios, but still include climate-carbon cycle interactions. The modelled land and ocean carbon sinks respond interactively to changes in climate and $CO_2$, such that the emissions compatible with each scenario can easily be diagnosed for each ESM[25]. The cumulative global fossil fuel $CO_2$ emissions ($E_{ff}$) compatible with each scenario are diagnosed from the change in atmospheric carbon content associated with the prescribed change in atmospheric $CO_2$ ($\Delta C_A$), and the simulated changes in global ocean carbon storage ($\Delta C_O$) and global land carbon storage ($\Delta C_L$):

$$E_{ff}(t) = \triangle C_A(t) + \triangle C_O(t) + \triangle C_L(t) \qquad (1)$$

In this study, we calculate the terms in this equation relative to 1850 (see Methods). In order to convert the cumulative fossil fuel emissions into the total cumulative emissions ($E$) we need to add on the cumulative global net land use emissions associated with each SSP scenario. Some ESMs now attempt to model net land-use change emissions interactively[26], but this is not yet routinely the case. We therefore follow the previous studies[8] in prescribing the standard net land-use changes ($E_{lu}$) associated with each SSP scenario[24]:

$$E(t) = \triangle C_A(t) + \triangle C_O(t) + \triangle C_L(t) + E_{lu}(t) \qquad (2)$$

We look for emergent relationships between the cumulative emissions calculated from Eq. (2), and global warming. As we want to constrain the policy-relevant carbon budgets for the Paris climate targets, we work in terms of the 'specific carbon budget' (i.e. the cumulative emissions per unit of global warming). This allows us to combine model differences and observational uncertainties in both global warming and cumulative emissions, into a single metric (see Methods for further details). Throughout this paper, unless otherwise stated, we quote cumulative carbon emissions from 1850, and mean values with likely ranges (66% confidence limits).

## Results

Figure 1 shows annual mean global mean values from nine ESMs for the ssp245 scenario. Even under this common prescribed concentration scenario, projections of global warming differ significantly between ESMs, ranging from 2 K to 4.5 K by 2100 (Fig. 1a). The primary reason for this is the large range in climate sensitivity to $CO_2$ across the CMIP6 model ensemble[9]. Projected changes in carbon storage show a smaller range across the models, especially for ocean carbon storage (Fig. 1b). When combined with the prescribed changes in atmospheric carbon

and estimated net land-use emissions (see Fig. S1), these changes in carbon storage can be used to diagnose the cumulative global $CO_2$ emissions using Eq. 2 (Fig. 1d). Similarly, the cumulative emissions consistent with other SSP scenarios (ssp126, ssp370, ssp585) can be diagnosed for each model (see Figs. S2, S3, S4).

Figure 2a shows approximately linear relationships between decadal mean cumulative global $CO_2$ emissions and global warming, for each of the SSP scenarios (symbols) and for each CMIP6 ESM (colours). The linearity is less clear below 0.5 °C because natural temperature variability has a relatively larger impact at these lower overall levels of warming, and because of the counteracting cooling effects of anthropogenic aerosols which were especially significant prior to 1980[21]. However, at larger values of global warming the CMIP6 models confirm the near proportionality between cumulative emissions and global warming which underlies the concept of the eTCRE and the notion of a carbon budget for each level of global warming[1,2,6,27,28]. Previous studies have attempted to explain the linearity between global warming and cumulative $CO_2$ emissions, either in terms of the compensating effects of either ocean heat and carbon uptake[29], or of carbon sink saturation and the logarithmic dependence of global warming on the $CO_2$ concentration[27,30]. In the context of this paper, we note that the linearity is clear in the latest CMIP6 models and focus on using that fact to derive emergent constraints on carbon budgets in the real world.

In Fig. 2a, the colours represent different models, while the symbols represent different scenarios. It is difficult to distinguish the scenarios because, for a given model, all scenarios lie on essentially the same line. In other words, the different lines are determined by the model rather than the scenario. The implication here is that uncertainties associated with eTCRE differences across models are much larger than the uncertainties associated with the non-$CO_2$ forcing factors (e.g. aerosols, trace GHGs) and net land-use emissions, all of which differ across the SSP scenarios[24]. This is a really important point as it is what allows us to constrain the carbon budget independently of the unknown future scenario.

Also shown by black symbols in Fig. 2a is the observed trajectory of cumulative emissions versus global warming based on the mean of four datasets of global mean temperature[31] and the cumulative emissions estimated by the Global Carbon Project (GCP)[11]. There are immediate hints here that the observations tend to favour a larger carbon budget per unit of global warming (and therefore a smaller TCRE) than most of the CMIP6 ESMs. This opens up the possibility of deriving Emergent Constraints[14,15] on the carbon budgets for each level of global warming.

We do not attempt to fit the slopes shown in Fig. 2a, but instead seek an emergent relationship between cumulative emissions and global warming up to the end of 2020, and the cumulative emissions at each future level of global warming. Figure 2b shows the emergent relationship between the 10-year mean specific carbon budget at 2 °C of global warming and the 10-year mean specific carbon budget up to the end of 2020, with each coloured symbol representing the results from a given model (colour) and scenario (symbol). The emergent relationship is very well defined ($r^2 = 0.92$) implying a clear relationship between the specific carbon budget up to the current day and the specific carbon budget at 2 °C. Figure 2b includes all of our SSP scenarios as we aim to find an emergent constraint on the future carbon budget, independent of the unknown future SSP scenarios. However, our emergent constraints are actually rather insensitive to scenario, as we can show by calculating emergent constraints for each SSP separately (see Table 1). We also find that our emergent constraints are robust to excluding any particular ESM, or all closely-related ESMs, from our analysis (see Table 2).

As is required to turn this emergent relationship into an emergent constraint, the x-axis variable can be estimated from observations[14,15]. Based on the mean of four global warming datasets[31] we calculate a

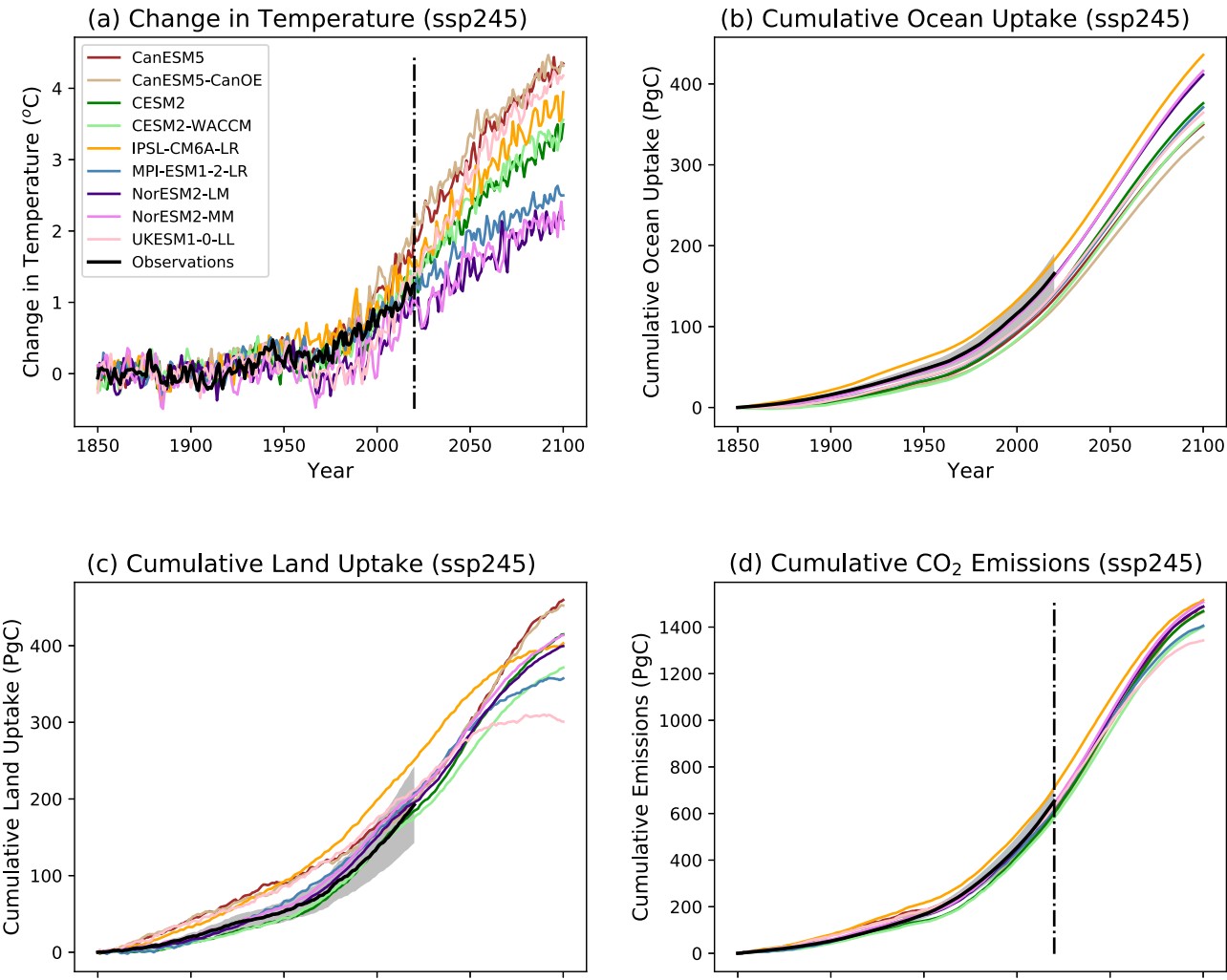

**Fig. 1 | Time-series for the historical period plus the SSP245 scenario, of annual global mean anomalies relative to the 1850–1899 mean.** (**a**) temperature change; (**b**) cumulative ocean carbon uptake; (**c**) cumulative land carbon uptake; (**d**) implied cumulative emissions (diagnosed using Eq. 1). Each coloured line represents a different CMIP6 Earth System Model (ESM) as identified in the key on (**a**).

The thick black line in (**a**) is the observational estimate from IPCC AR6 WG1 Chapter 2[31]. The thick black lines in (**b**–**d**) represent observational estimates from the Global Carbon Project[11], with grey bands showing the estimated standard error in these values (see Methods for details).

global surface air temperature (GSAT) anomaly of 1.09 °C for 2011 to 2020 compared to the 1850–1899 mean, with an assumed standard error of 0.12 °C (see Methods). Similarly, using data available from the Global Carbon Project[11] we calculate mean cumulative emissions of 604 PgC for 2011-2020 with an estimated uncertainty of ±10% (ref. 8). Together these observations therefore imply an observational estimate on the specific carbon budget up to the current day of 553±82 PgC/°C (see Methods), which is shown by the light-blue vertical bar in Fig. 2b.

The emergent relationship dotted black line and grey uncertainty bounds in Fig. 2b and this observational constraint can now be combined to provide an emergent constraint on the *y*-axis variable, using the technique used in many previous studies[17–19,32–34]. Figure 3 shows the resulting emergent constraint on the specific carbon budget for 2 °C of global warming. Figure 3a shows the probability density function (pdf) based on an equal weighting of the modelled values (grey histogram and black line representing a Gaussian fit to this histogram), along with the posterior distribution after applying the observational constraint (red line). The estimated value of the specific carbon budget for 2 °C moves upwards from 456 ± 123 PgC/°C in the equal-weighted model mean, to 524 ± 84 PgC/°C as a result of the emergent constraint,

implying a mean carbon budget for 2 °C of 1048 ± 167 PgC (Table 3). Figure 3b shows the corresponding cumulative distribution function (cdf) which is consistent with a carbon budget of 976 PgC for a 66% chance of avoiding 2 °C of global warming (compared to 804 PgC based on the equal-weighted model mean).

The resulting emergent constraint on the specific carbon budget to 2 °C is similar to the estimated specific carbon budget up to the current day based on observations[23], which is consistent with a near-constant specific carbon budget (and its inverse, eTCRE) up to 2 °C. However, a major advantage of our emergent constraint approach is that it does not require an implicit assumption of a constant specific carbon budget to higher levels of global warming, or that it will be independent of the details of the ssp scenario that is followed. Instead we can repeat the emergent constraint analysis shown in Fig. 3 for other levels of global warming (Table 3, Table S1), in each case deriving constraints on the carbon budget from the emergent relationship across the models between the current specific carbon budget and the specific carbon budget at the chosen level of global warming (see Figs. S5, S6 for further examples for 1.5 K and 3 K of global warming). As results are taken from four different ssp scenarios (ssp126, ssp245, ssp370, ssp575), a range of possible future non-CO$_2$ factors (trace

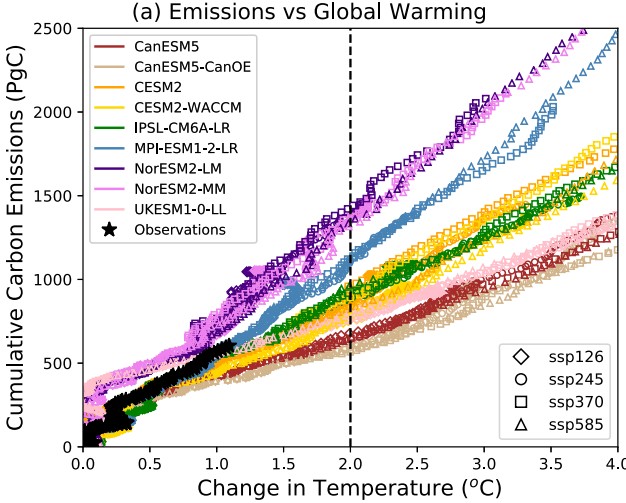

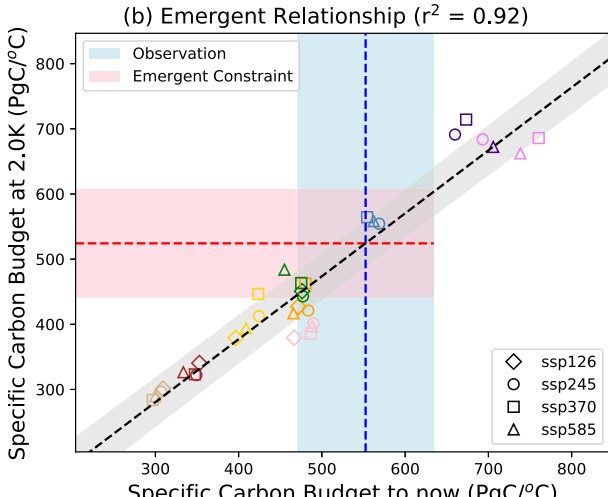

**Fig. 2 | Relationship between cumulative emissions and global warming, for the historical simulations plus four different SSP scenarios (ssp128, ssp245, ssp370, ssp585).** Panel (**a**) plots cumulative emissions since 1850 (diagnosed using Eq. 1) against global warming since 1850, for each of the nine CMIP6 ESMs (coloured symbols) and the Global Carbon Project (GCP) plus global mean temperature observations (black stars). The values shown here are 10 year centred means to minimise the impact of interannual variability. The different symbols denote mean values taken from each of the SSP scenarios. Panel (**b**) shows the emergent relationship between the specific carbon budget (in PgC/°C) for 2 °C of global warming, and the specific carbon budget to 'now' (based on the 10-year means for 2011 to 2020). The vertical blue dashed line shows the observational estimate of the latter, with the light-blue bar showing an estimate of the (66%) uncertainty in this estimate. The horizontal pink bar shows the resulting emergent constraint on the specific carbon budget for 2 °C of global warming, with the red dashed line indicating the central estimate.

greenhouse gases, aerosols, and land-use change) are included within each emergent constraint.

Figure 4 shows these emergent constraints on the carbon budget as a function of global warming. Despite many modelled non-linearities and tipping points in the climate system, the estimated carbon budget shows a linearity with global warming at least as far as 4 °C (Fig. 4a). The ordinary-least-squares gradient from 1.2 °C of global warming onwards (red dashed line in Fig. 4a) defines a specific carbon budget of 473 ± 80 PgC/°C which is markedly lower than the specific carbon budget to the current day (553 ± 82 PgC/°C). Figure 4b shows how the specific carbon budget varies with global warming. The specific carbon budget would be a flat line against global warming if global warming was strictly proportional to cumulative emissions (i.e. linear and with zero intercept). The specific carbon budget is however higher for lower values of global warming because of the cooling effect of anthropogenic aerosols in the second half of the 20th century. Our specific carbon budget for future warming of 473 ± 80 PgC/°C, is equivalent to a likely range for the effective Transient Climate Response to Emissions (eTCRE) of 1.8 to 2.6 °C per 1000 PgC, with a mean estimate of 2.1 °C per 1000 PgC.

We have focussed our study on the CMIP6 ESMs, because (a) these are most recent generation of models that have undertaken the required model runs; (b) in CMIP6 many more models completed all of the relevant runs for ssp126, ssp245, ssp370, ssp585, which allows us to

cleanly account for the effects of different future scenarios of non-$CO_2$ factors. Nevertheless, to test the robustness of our findings we repeated our emergent constraints using climate-carbon cycle runs from the previous CMIP5 generation of models. Figure S7 in the Supplementary Information shows the equivalent emergent constraint on the carbon budget for 2 °C based on the available CMIP5 model runs. Although the emergent relationship for the CMIP5 models is weaker (in part because it is based on 8 data points rather than 36), it leads to a very similar best estimate for the carbon budgets (e.g. for 2 °C of global warming: 1045 PgC from CMIP5; 1048 PgC from CMIP6).

For more direct comparison to the result provided in Table 5.8 of WG1 of the Intergovernmental Panel on Climate Change, 6th Assessment Report[8], we have also developed an emergent constraint on the remaining carbon budgets from 2020 onwards. The remaining carbon budget for each model is first calculated by subtracting the previously calculated carbon budget for 'now' (based on 10 year means from 2011 to 2020) from the carbon budget for the given level of global warming. This actually gives the remaining carbon budget compared to the mean carbon budget over the period 2011–2020, so we need to make a correction to provide an estimate of the remaining carbon budget from the beginning of 2020 onwards, for comparison to the IPCC AR6. We do this using the GCP data[11] which indicates that the cumulative emissions to the end of 2019 were 642 PgC, as opposed to 604 PgC for the mean over the period 2011–2020. We therefore apply a correction to the remaining carbon budgets from the models of 604 − 642 = −38PgC, to account for cumulative emissions from the centre of the last 10-year meaning period to the end of 2019.

Figure 5a shows a strong emergent relationship between this remaining carbon budget for 2 °C of global warming (from 2020 onwards) and the specific carbon budget up to 'now'. We find similar emergent relationships for other levels of global warming, as shown in Fig. 5b. The remaining carbon budgets from this analysis are within the likely bounds of the IPCC AR6, but with slightly higher central estimates in this study, and with much tighter constraints on the upper likely bound (see Table S1). Our mean estimates of the carbon budget for the Paris Targets of 1.5 °C and 2 °C are respectively about 46 and 52

**Table 1 | Scenario sensitivity of emergent constraint on the specific carbon for 1.5 °C of global warming**

| Scenario | Emergent constraint on specific carbon budget (PgC/°C) |
|---|---|
| ssp126 | 554 [464, 649] |
| ssp245 | 537 [456, 619] |
| ssp370 | 552 [465, 641] |
| ssp585 | 523 [455, 592] |
| *All scenarios* | 541 [461, 622] |

The bracketed values give the 'likely' bounds (66% confidence limits).

PgC higher, which is equivalent to about 4 years of global anthropogenic emissions at the current rate.

## Discussion

We have demonstrated top-down emergent constraints on cumulative carbon budgets for different levels of global warming. This approach makes use of an emergent relationship across an ensemble of CMIP6 Earth System Models (ESMs), between the modelled carbon budgets at each level of global warming and the simulated carbon budget up to the current day. As the emergent relationship includes projections based on different Shared Socioeconomic Pathways, it also folds in uncertainties associated with different non-CO₂ factors (such as trace greenhouse gases, aerosols and land-use). Using this approach we estimate carbon budgets for the Paris targets of 812 [691, 933] PgC for 1.5 °C of global warming and 1048 [881, 1216] PgC for 2 °C of global warming, which are 86 PgC and 137 PgC larger than estimates based on the ensemble ESM mean. We infer a reduced mean specific carbon budget in the future (473 ± 80 PgC/°C beyond 1.2 °C of global warming) compared to the past (553 ± 82 PgC/°C), primarily due to the decline in cooling atmospheric aerosols. However, the linearity

between future cumulative emissions and future global warming is found to be maintained at least until 4 °C of global warming, and is consistent with an effective Transient Climate Response to Emissions (eTCRE) of 2.1 [1.8, 2.6] °C/1000PgC, from a global warming of 1.2 °C onwards. We have also presented additional emergent constraints which imply remaining carbon budgets from 2020 onwards of 186 [69, 304] PgC for 1.5 °C and 422 [258, 586] PgC for 2 °C. These values are about 50 PgC larger, but within the likely bounds, of those given in the IPCC 6th Assessment, Working Group 1 Report[8], but do not require potentially inconsistent assumptions about the likely range of a number of climate and carbon cycle sensitivity factors, or changes in non-CO₂ forcing factors. At current rates of global CO₂ emissions of approximately 11 PgC yr⁻¹, our analysis suggests that the remaining carbon budgets will be used-up by the year 2037 [2026, 2048] for 1.5 °C, and by 2058 [2043, 2073] for 2 °C.

## Methods

In our analysis, we included CMIP6 models that provided the necessary land-atmosphere and ocean-atmosphere CO₂ fluxes for the historical run (1850–2014) and for all of the ssp126, ssp245, ssp370, ssp585 scenario runs (2015-2100). In addition, we required that each model provided a land-sea mask to enable global means to be accurately calculated. To be consistent with the precribed CO₂ runs that we use for the ssp scenarios, we use the CMIP6 historical runs (with prescribed atmospheric CO₂) rather than the CMIP6 esm-hist runs (which calculate the atmospheric CO₂ interactively based-on prescribed CO₂ emissions). Time-series of global annual mean near surface air temperature (CMIP variable: tas), Net Biome Productivity over land (CMIP variable: nbp), and ocean CO₂ flux (CMIP variable: fgco2), were calculated from monthly spatial fields downloaded via the ESGF node (https://esgf-index1.cdea.ac.uk/projects/cmip6-ceda/) on 16 October 2020.

We calculate temperature anomalies relative to the mean for 1850–1899 inclusive (as in the IPCC AR6[31]):

$$\triangle T(N) = T(N) - \frac{1}{50} \sum_{y=1850}^{y=1899} T(y) \qquad (3)$$

where $T(N)$ is the global mean, annual mean temperature for year $N$. For the models, we calculate the carbon store changes (PgC) for land

**Table 2 | Sensitivity of the emergent constraint on the specific carbon for 1.5 °C of global warming, to leaving specific models out of the emergent relationship**

| Model(s) left out | Emergent constraint on specific carbon budget (PgC/°C) |
|---|---|
| CESM2 | 548 [465, 631] |
| CESM2-WACCM | 545 [462, 629] |
| CanESM5 | 544 [459, 629] |
| CanESM5-CanOE | 543 [458, 630] |
| IPSL-CM6A-LR | 545 [461, 629] |
| MPI-ESM1-2-LR | 539 [459, 620] |
| NorESM2-LM | 532 [458, 606] |
| NorESM2-MM | 556 [468, 644] |
| UKESM1-0-LL | 550 [468, 632] |
| CESM2-WACCM; CanESM5-CanOE; NorESM2-MM | 556 [460, 650] |
| *None excluded* | 541 [461, 622] |

The bracketed values give the 'likely' bounds (66% confidence limits).

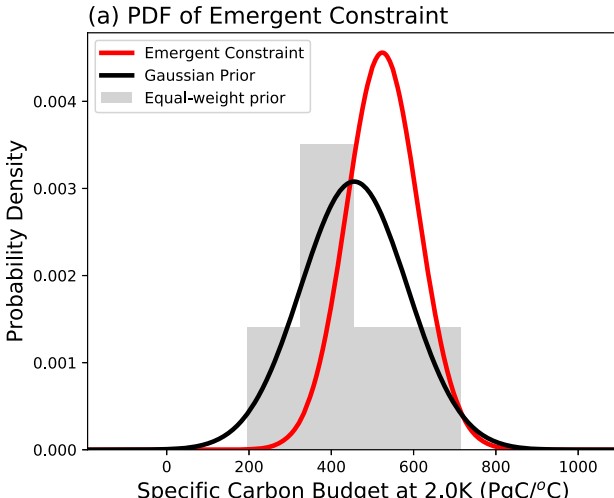

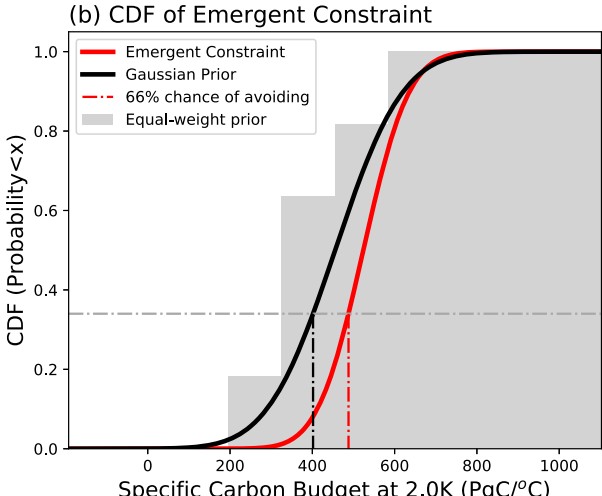

**Fig. 3 | Emergent constraint on the specific carbon budget for 2 °C of global warming. a** probability density function (PDF). **b** Cumulative distribution function (CDF). The grey histograms show the distributions derived from the equal-weighted raw model output, and the black line is a Gaussian with the same mean and standard deviation. The thick red line shows the emergent constraint on the distribution.

and ocean, for each year $N$, as the sum of the annual mean fluxes:

$$\triangle C_O(N) = \sum_{y=1850}^{y=N} fgco2(y) \qquad (4)$$

$$\triangle C_L(N) = \sum_{y=1850}^{y=N} nbp(y) \qquad (5)$$

where $fgco2(y)$ is the annual mean, global total ocean $CO_2$ flux (PgC yr$^{-1}$), and $nbp(y)$ is the annual mean, global total net ecosystem productivity over land (PgC yr$^{-1}$). The equivalent prescribed increase in atmospheric carbon is calculated as:

$$\triangle C_A(N) = 2.12\{co2(N) - co2(1850)\} \qquad (6)$$

where $co2(N)$ is the annual mean, global mean $CO_2$ concentration for year N in ppmv, and 2.12 is the usual conversion factor from ppmv to PgC. We then use Eq. 2, with the prescribed cumulative net land-use emissions $E_{lu}(N)$ (see Figure S1) to calculate the total cumulative emissions for each year $E(N)$. To minimise the impact of interannual

### Table 3 | Emergent constraints on cumulative carbon budgets, and remaining carbon budgets from the beginning of 2020, as a function of global warming

| Global Warming (K) | Carbon budget (PgC) | Remaining carbon budget from 2020 (PgC) |
|---|---|---|
| 1.3 | 729 [621, 838] | 103 [0, 207] |
| 1.4 | 772 [660, 884] | 147 [36, 259] |
| 1.5 | 812 [691, 933] | 186 [69, 304] |
| 2.0 | 1048 [881, 1216] | 422 [258, 586] |
| 2.5 | 1272 [1065, 1480] | 645 [440, 854] |
| 3.0 | 1520 [1274, 1774] | 894 [648, 1146] |
| 3.5 | 1750 [1457, 2053] | 1122 [828, 1427] |
| 4.0 | 2023 [1663, 2400] | 1387 [1030, 1761] |

The square brackets give the 'likely' bounds (66% confidence limits).

variability, we calculate centred 10 year means of $\triangle T(N)$ and $E(N)$, which we use for both model and observations within our emergent constraints. We use observational data up to 2020, this implies that our observational constraint is based on 10 year means from 2011 to 2020 inclusive, and is therefore centred on the end of 2015.

We use the data plotted in Chapter 2 of the IPCC AR6 WG1[31], Figure 2.11a (https://data.ceda.ac.uk/badc/ar6_wg1/data/ch_02/ch2_fig11/v20211207/Figure2_11_panel_a.csv), which represents the mean of four datasets corrected for the difference between Global Mean Surface Temperature (GMST) and Global Surface Air Temperature (GSAT). From these data, we calculate a mean anomaly in GSAT of 1.093 °C for 2011 to 2020 inclusive, with an assumed standard error of 0.12 °C. The figure of 0.12 °C comes from IPCC AR6 WG1, Chapter 2 (Cross Chapter Box 2.3, Table 3, Footnote b) that states a 'likely uncertainty range of ±0.12 °C' for the decadal mean global warming relative to the 1850–1900. Observational estimates of the mean cumulative emissions for 2011–2020 are calculated from the the Global Carbon Project[11] as 604.0 PgC with an estimated uncertainty of ±10%. This estimate of uncertainty in the historical cumulative fossil fuel emissions is based on Chapter 5 of the IPCC AR6 WG1 report[8] (see Table 5.8, footnote b). Based on these factors the central estimate of the observed specific carbon budget is therefore 604.0/1.093 = 552.5 PgC/°C. In the absence of detailed information on the distributions of the random errors in the cumulative emissions and the decadal mean temperature anomalies, we assume that the uncertainty in the observed specific carbon budget is Gaussian distributed. We estimate the fractional uncertainty in this specific carbon budget ($f_{scb}$) by combining the fractional uncertainties in quadrature, of the cumulative carbon emissions ($f_{ce} = 0.1$), and the global temperature anomaly ($f_{dt} = 0.12/1.09 = 0.11$):

$$f_{scb} = \sqrt{f_{ce}^2 + f_{dt}^2} = 0.149 \qquad (7)$$

We therefore derive an observational constraint on the specific carbon budget of 553 ± 82 PgC/°C.

The emergent constraints derived in this study use a very similar approach to a number of previous studies[17–19,32–34], applying an ordinary-least-squares (OLS) fit between the predictand and the predictor variable. More sophisticated Bayesian approaches have been proposed and applied in other studies[35–37]. However, these approaches

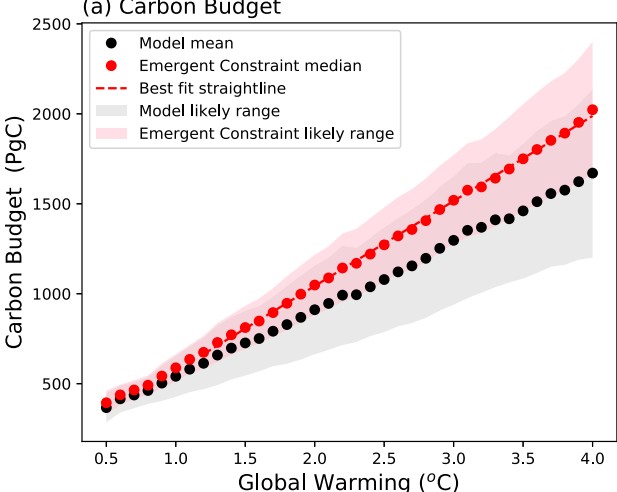

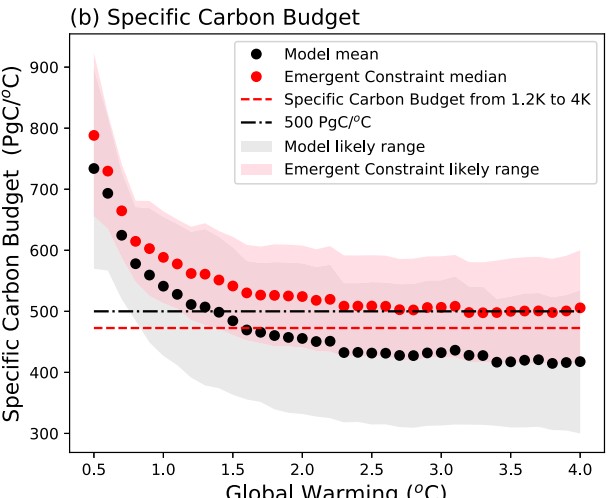

**Fig. 4 | Estimates of the carbon budgets consistent with different levels of global warming.** a Emergent constraints on the carbon budgets (red dots) compared to the ensemble model mean (black dots), where the red dashed line is a ordinary-least-squares straightline from 1.2 °C onwards with a gradient of 473 PgC/°C. b Specific carbon budgets (PgC/°C) consistent with the data shown in (a), where the red dashed line shows the mean specific carbon budget from 1.2 °C onwards

(473 PgC/°C) and the black dot-dash line shows a mean specific carbon budget of 500 PgC/°C[1]. The red dots and pink (66%) uncertainty bounds are the emergent constraints derived in this study. Similarly, the central estimates and 66% uncertainty bounds are shown by black dots and grey shading for the equal-weighted model ensemble.

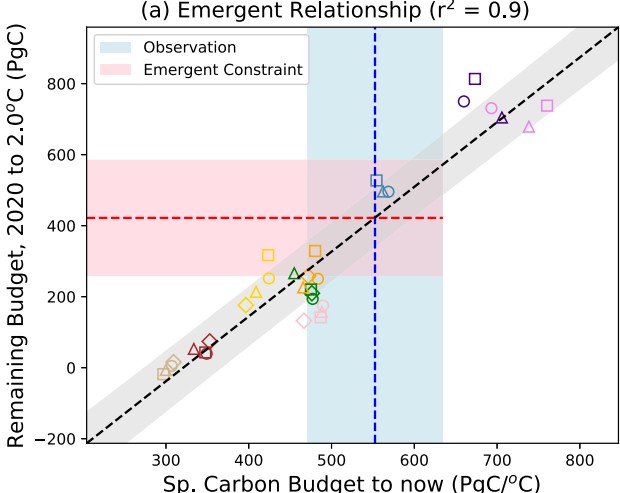

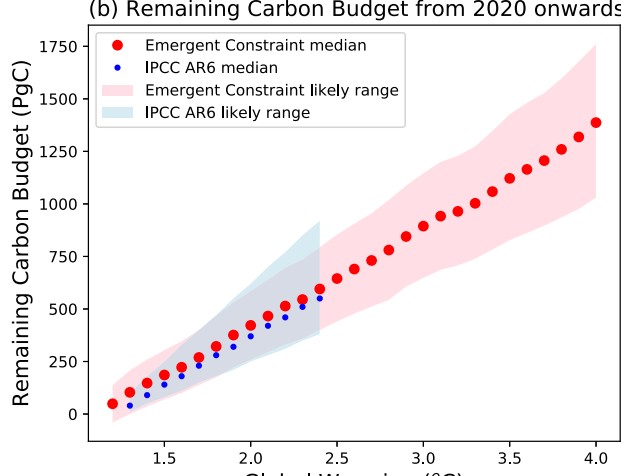

**Fig. 5 | Emergent constraint on the remaining carbon budget for different levels of global warming. a** Emergent relationship between the remaining carbon budget from 2020 onwards to 2 °C of global warming, and the specific carbon budget to 'now' (based on the 10-year means for 2011 to 2020). The vertical blue dashed line shows the observational estimate of the latter, with the light-blue bar showing an estimate of the (66%) uncertainty in this estimate. The horizontal pink bar shows the resulting emergent constraint on the remaining carbon budget for 2 °C of global warming, with the red dashed line indicating the central estimate. **b** Remaining carbon budget as a function of global warming, for comparison with estimates from the IPCC AR6[8] (blue dots). The central estimates and 66% uncertainty bounds are shown by red dots and pink shading for the emergent constraints derived in this study, and by blue dots and light-blue shading for the estimates from the IPCC AR6.

have been found to yield very similar constraints to OLS when the emergent relationships are strong and linear[21], as they are in this study.

We make use of emergent relationships across the CMIP6 models between the specific carbon budget at a given level of global warming and the specific carbon budget up to the current day. The probability density of the y-axis variable, $P(y)$, is given by

$$P(y) = \int_{-\infty}^{\infty} P\{y|x\} P(x) \, dx \qquad (8)$$

Where the probability of y given x, $P\{y|x\}$, is derived from the best fit emergent relationship, and the probablity of x, $P(x)$, is the observational constraint. The emergent constraint $P(y)$ is therefore affected by both the quality of the emergent relationship $P(y|x)$ and the uncertainty in the observational constraint $P(x)$. We integrate this equation numerically to derive the constraint on the y variable, $P(y)$, making use of the estimated uncertainty in the observational constraint and assuming, in this case, that $P(x)$ can be represented by a Gaussian distribution. In this study, the emergent relationships are very well approximated by a linear regression, so that

$$P\{y|x\} = \frac{1}{\sqrt{2\pi\sigma_f^2}} \exp\left\{ -\frac{(y - f(x))^2}{2\sigma_f^2} \right\} \qquad (9)$$

where $f(x)$ is the linear regression, calculated in this study using ordinary least squares, and the x-dependent prediction error of the regression is given by

$$\sigma_f(x) = s \sqrt{1 + \frac{1}{N} + \frac{\{x - \bar{x}\}^2}{N\sigma_x^2}} \qquad (10)$$

and $N$ is the number of points, $\bar{x}$ and $\sigma_x^2$ are the mean and variance of the x-axis variable, and $s$ is the standard error in the fit, given by the square root of

$$s^2 = \frac{1}{(N-2)} \sum_{n=1}^{N} \{y_n - f_n\}^2 \qquad (11)$$

## Data availability
Data sets generated during the current study are available via Figshare (https://doi.org/10.6084/m9.figshare.24960540).

## Code availability
The python code used for the analysis presented here is available via Figshare (https://doi.org/10.6084/m9.figshare.24960540).

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

## Acknowledgements

The authors acknowledge funding from the ERC ECCLES project grant no. 742472 (P.C., M.W.), the EU 4C project grant no. 821003 (P.F., P.C., R.V.), the EU ESM2025 project grant no. 101003536 (C.J., J.R., P.F.), the Joint UK BEIS/Defra Met Office Hadley Centre Climate Programme, GA01101 (C.J.), and a Marie Sklodowaka-Curie Fellowship grant no. 101020078 (N.R.). We also acknowledge the World Climate Research Programme's Working Group on Coupled Modelling, which is responsible for CMIP, and we thank the climate modelling groups for producing and making available model output from the CMIP6 models.

## Author contributions

P.M.C. led the study and drafted the manuscript. M.W. downloaded and processed the CMIP6 model outputs to produce the required time-series data. P.F. provided guidance on an early draft of the study, and advised on the relationship to previous observational constraints on carbon budgets. C.J. and J.R. provided information and context with regard to the carbon budgets estimated in the IPCC AR6 WG1 report. N.R. and R.V assisted with the revisions required to the original submission, and checked the statistical method.

## Competing interests

The authors declare no competing interests.
