## [Peer Review File · Nature Communications]

Emergent constraints on carbon budgets as a function of global warmingReviewers' Comments:

Reviewer #1 (Remarks to the Author):

The authors present new work using emergent constraints to reduce uncertainty in carbon budgets as a function of global warming. The research provides an advance on the literature and should be of interest to the broader community.

The current draft, however, reads more like an n-1 or n-2 version and therefore doesn't reflect the polished efforts needed for Nature Comm. There are a number of technical issues, missing references, and areas where additional clarity would benefit the manuscript.

I do not think these are insurmountable but will require additional effort. Based upon these major revisions, the paper could become suitable for publication.

Reviewer #2 (Remarks to the Author):

Review of "Emergent constraints on carbon budgets as a function of global warming" by Cox et al., submitted to Nature Communications.

Manuscript NCOMMS-22-50851

The authors introduce emergent constraints on the carbon budget as a function of global warming. By establishing a linear, emergent relationship between the specific carbon budget at a given level of global warming (y), and the specific carbon budget to "now" (x), the authors use an observational constraint to predict the cumulative carbon budget, with its uncertainty quantified. Levels of global warming between 1 °C and 4 °C are considered, from which the remaining carbon in the budget beyond "now" (2020) can be predicted.

Motivation for this research is to reduce uncertainty in the latest CMIP6 Earth Systems Models projections. The authors find an emergent constraint that reduces uncertainties, which has the potential to contribute to climate-change policy-making. However, we have concerns about the statistical methodology used. This brings into question the veracity of the prediction and uncertainty calculations, both of which may affect the final results and yield incorrect conclusions.

General comments

Statistical modeling of the emergent relationship and constraint is central to the manuscript. The authors make clear that there is uncertainty not only in y but also in x , which they specify through $P(x)$. As a

consequence, the slope estimates obtained from Figure 2a are biased. There are papers that deal with this, the original one due to Bowman et al. (2018) who present a solution using a hierarchical statistical modeling framework. Hierarchical emergent constraint (HEC) modeling accounts for error in the x-variable and yields accurate and precise estimates, unlike those in Figure 2b. This framework has been applied in a number of articles; see, for example, Thackeray and Hall (2019), Barkhordarian et al. (2021), Shiogama et al. (2022), and Chen et al. (2022). The submitted manuscript needs to fit the emergent relationship based on HEC because the x-variable comes with uncertainty.

We give more specific comments, by section, below.

Introduction

- While the introduction is about remaining carbon budgets, most of the rest of the manuscript concentrates on the cumulative carbon budget (see conclusions and Table 1). The discussion around Figure 4c on lines 161-166 shows what remains as a function of global warming, and we suggest those results also be included in Table 1. The 30 PgC difference from IPCC6 (lines 179-180) is not significant. An emergent-constraint analysis needs to recognize properly the uncertainty in both y and x in order to provide policy-makers with accurate (unbiased) and precise (smallest uncertainty) results.

Method

- The proposed method consists of several components: the modeled carbon budgets, a linear, emergent relationship, and the observational constraint. While discussion for the first one is well presented, those for the other two are quite limited. The linear emergent relationship deserves at least an equation, and it should be shown clearly how it is used to obtain the emergent constraint along with its uncertainty.
- Please provide a clear definition of the observational constraint. Throughout the paper, it is either presented as a central estimate +/- errors, or a probability density function of the predictor in the Appendix. Sometimes the error bounds are not symmetric (e.g., Figure 4c); however, the +/- results given in the text assume they are symmetric.

Results

- In Bayesian statistics, the prior and posterior are usually distributions on the same variable, the former incorporating prior information and the latter being an update given new information. In fact, the statistical model in this manuscript is on the bivariate quantities (x, y), where $P(x, y) = P(y | x)P(x)$, and interest is in the marginal distribution, $P(y) = \int P(x, y)dx$. This is not an analogue of Bayes theorem (line 356). The theorem involves the inverse probability $P(x | y)$, which is not of great interest here for policy-making. The authors use “prior” for $P(x)$ in Figure 3, which will cause confusion for readers.

- Why did the authors use median instead of mean for the results based on emergent constraints? The uncertainty calculations for medians is different than that for means; they cannot be used interchangeably, since the sample median has a different standard error than that of the sample mean.

Conclusions

- The remaining carbon budget will be “spent” over time. In the 30 years between 2020 and 2050 (for example), what is the target spend-rate for 1.5 °C, 2.0 °C, and 2.5 °C? How does that compare to the spend-rates up to 2020, where uncertainties should be incorporated into the comparison? These might be questions to address using the emergent-constraint analysis.

Appendix

- Uncertainty calculation for specific carbon budget (line 346). Using an additive formula to calculate uncertainty is suitable when one wants to compute $\text{var}(U + V)$. However, it is not true that

$$\frac{\text{var}(U + V)}{[E(U + V)]^2} = \frac{\text{var}(U)}{[E(U)]^2} + \frac{\text{var}(V)}{[E(V)]^2}.$$

This is in effect what the authors are proposing on line 346; one has to be careful when combining relative uncertainties.

- The formulas for the uncertainty $\sigma_f(x)$ on lines 365–374 assume that x is known with certainty. However, x has uncertainty, and the authors estimate the uncertainty on y taking into account the uncertainty on x , as specified on lines 360–363. What then is the relationship between what is presented on lines 360–363 and lines 365–374? The two derivations do not result in the same uncertainty measure. This is another example of the need to use HEC from the beginning, where the emergent-relationship model has uncertainty in x , and every formula takes that into account.
- Uncertainty calculation for the remaining carbon budget. Notice that $\text{remain} = \text{total} - \text{hist}$ or $r = y - x$. The authors claim that r and x are independent, which is almost certainly not true; since the total y is constrained for a given level of global warming, r and x will be negatively correlated. The appropriate calculation is

$$\sigma_r^2 = \sigma_y^2 + \sigma_x^2 - 2\rho_{xy}\sigma_x\sigma_y,$$

where σ^2y and σ^2x are the variances of y and x , and ρ_{xy} is the correlation between x and y . In the manuscript, $\epsilon^2_{\text{total}}$ and ϵ^2_{hist} are estimates of σ^2y and σ^2x , respectively. The authors still need to estimate the third term.

Minor comments

- Line 100: “based-on” → based on
- Line 155: “best-fit” → ordinary-least-squares?
- Line 168: “K” → °C (and throughout the manuscript)
- Line 301: “Timeseries” → Time series
- Lines 319, 328: “annual mean, global mean” → annual mean, global total? (Note: The unit of flux is “PgC/time/area”. To obtain the unit of “PgC/time” that the authors use, a total over “area” is needed. This applies to global, land, ocean, or any region of interest.

References

- Barkhordarian, A., Bowman, K. W., Cressie, N., Jewell, J., and Liu, J. (2021), “Emergent constraints on tropical atmospheric aridity—carbon feedbacks and the future of carbon sequestration,” *Environmental Research Letters*, 16, 114008.
- Bowman, K. W., Cressie, N., Qu, X., and Hall, A. (2018), “A hierarchical statistical framework for emergent constraints: Application to snow-albedo feedback,” *Geophysical Research Letters*, 45, 13–050.
- Chen, Z., Zhou, T., Chen, X., Zhang, W., Zhang, L., Wu, M., and Zou, L. (2022), “Observationally constrained projection of Afro-Asian monsoon precipitation,” *Nature Communications*, 13, 2552.
- Shiogama, H., Watanabe, M., Kim, H., and Hirota, N. (2022), “Emergent constraints on future precipitation changes,” *Nature*, 602, 612–616.
- Thackeray, C. W. and Hall, A. (2019), “An emergent constraint on future Arctic sea-ice albedo feedback,” *Nature Climate Change*, 9, 972–978.

Response to Reviewer 1

We thank the reviewer for their detailed comments on our original submission. It has taken us some time to answer these, but we believe that they stimulated significant improvement in our revised paper, as outlined below. Reviewer comments are shown in normal black text, and our responses are in dark blue italics.

2-1 This notation is confusing. The cumulative emissions are described on the left-hand side of the equation, but as E_{ff} where the right hand side is a change against a baseline ΔC . To be consistent, it should be $\Delta C_{\{ff\}}$. Also, I don't think this is absolute time, t , but rather a change in time, Δt , relative to a starting point (presumably 1850 or thereabouts).

Response: *We have added a sentence to justify our notation: “The ΔC terms relate specifically to changes in modelled carbon stores, while E_{ff} represents the diagnosed cumulative emissions from fossil fuels (i.e. an external factor)”. The carbon balance (Eqn 1) applies at all times t , but we have added the following text to clarify the point raised by the reviewer: “In this study we calculate the terms in this equation relative to 1850 (see Methods)”.*

2-2 Why concentration and not emission scenarios? If you're using concentration scenarios then the land and ocean can't really interact properly. The rationale and caveats of this assumption need to be addressed in the paper. For a given emission scenarios, one can test to what extent the right hand side matches the left-hand side in Eq. 1. That should account for coupling.

Response: *There is a long history of doing things this way, and tests to date have shown that runs with prescribed emissions and prescribed concentrations lead to similar climate-carbon cycle feedbacks (Hibbard et al., 2007). We have added the following text to clarify: “This approach was adopted in CMIP5 and CMIP6 as it allows ESMs to use comparable concentration scenarios, but still to include climate-carbon cycle interactions. The modelled land and ocean carbon sinks still respond interactively to changes in climate and CO_2 , such that the emissions compatible with each scenario can easily be diagnosed for each ESM (Hibbard et al., 2007)”.*

2-3 All carbon: CO_2 , CH_4 , CO , COS , etc? I doubt it. Clarify. Also, I think cumulative atmospheric carbon “content” is what is meant here.

Response: *The carbon content referred to here is indeed associated with CO_2 . This is by far the dominant component of atmospheric carbon in terms of carbon mass, but point taken. For clarity, the text “prescribed change in atmospheric carbon content” has therefore been replaced with “change in atmospheric carbon content associated with the prescribed change in atmospheric CO_2 ”.*

2-4 A prescribed change of what with respect to what baseline. Provide more context for the reader.

Response: *All cumulative carbon fluxes are calculated from 1850, as stated in ‘Methods’ under ‘b) Data Processing’. However, to improve the clarity of the main text we have added the following sentence after Eqn 1: “In this study we calculate the terms in this equation relative to 1850 (see Methods)”.*

2-5 I know this is frequently done, but just adding linearly the land-use emissions to the natural land fluxes ignores the coupling between the two. If it holds then under an emissions-driven scenario, $E_{LUC} = C_a - E_{FF} - E_{ocean} - E_{land}$. Please verify. If not the same, the please explain why and add appropriate caveats in the text.

Response: *This is indeed how things are routinely done in the estimates of carbon budgets from current Earth System Models. The equation that the reviewer writes down is a statement of carbon conservation if ‘ E_{LUC} ’ represents net emissions from land-use change (i.e. emissions from anthropogenic land—use change minus carbon accumulated by regrowth on abandoned lands). It*

therefore applies to emission-driven runs as well as to concentration-driven runs. We have added the following text to explain the current situation: ***“Some ESMs now attempt to model net land-use change emissions interactively (Lawrence et al., 2016), but this is not yet routinely the case. We therefore follow previous studies (Canadell et al., 2021) in prescribing the standard net land-use changes (E_{lu}) associated with each SSP scenario (Meinshausen et al., 2020)”***

2-6 It's not approximately linear at all for temperature changes less than 0.5C. Explain.

Response: The following text has been added: “The linearity is less clear below 0.5°C because natural temperature variability has a relatively larger impact at these lower overall levels of warming, and because of the counteracting cooling effects of anthropogenic aerosols which were especially significant prior to 1980 (Nijse et al., 2020)”

2-7 It's not intuitively obvious why this is the case given carbon and climate feedbacks. Please add a sentence or two providing some context.

Response: This study is focussed on deriving constraints on carbon budgets, but as requested we have added some text and two references on the linearity between cumulative emissions and global warming: “Previous studies have attempted to explain the linearity between global warming and cumulative CO₂ emissions, either in terms of the compensating effects of either ocean heat and carbon uptake (Goodwin et al., 2015), or of carbon sink saturation and the logarithmic dependence of global warming on the CO₂ concentration (MacDougall and Friedlingsten, 2015; MacDougall, 2016). In the context of this paper, we note that the linearity is clear in the latest CMIP6 models and focus on using that fact to derive emergent constraints on carbon budgets in the real world”

2-8 The “far exceed” is true for temperature changes exceeding 1.8C or so. That's not true for the nearfuture. So, for results out to 2100, that should be fine. Please add appropriate caveat.

Response: We suspect the reviewer may have misunderstood Figure 2a, which suggests that we have not explained it properly. To make it clearly we have replaced the two sentences beginning on line 93 with: “In Figure 2a, the colours represent different models, while the symbols represent different scenarios. It is difficult to distinguish the scenarios because, for a given model, all scenarios lie on essentially the same line. In other words, the different lines are determined by the model rather than the scenario. The implication here is that uncertainties associated with TCRE differences across models are much larger than the uncertainties associated with the non-CO₂ forcing factors (e.g. aerosols, trace GHGs) and net land-use emissions, all of which differ across the SSP scenarios²⁵. This is a really important point as it is what allows us to constrain the carbon budget independently of the unknown future scenario”

2-9 In addition to the authors, there are a number of other important references that should be included:

Thackeray et al, 2022 Nat. Climate Change
Shiogama et al, 2022, Nature
Barkhordarian et al, 2021, Env. Research Letters
Sansom et al, 2021, J. American Statistical Association.
Bretherton and Caldwell, 2020, J. Climate
Renoult et al, 2020, Climate of the Past
Including the carbon cycle and climate sensitivity not referenced.

Response We have included some references from this list that are pertinent to our revisions.

3-1 That's a bit colloquial, so technically it's not true. It's up to 2020. Say that.

Response: As requested, we have replaced “up to the current day” with “up to the end of 2020”.

3-2 Not clear what you mean by “specific” carbon budget vs just a carbon budget. Please clarify in text.

Response: *In the submitted manuscript we wrote:*

“We look for an emergent relationship between the specific carbon budget (cumulative emissions per unit of global warming) up to the current day and the specific carbon budget at each level of global warming. Perfect linear relationships passing through the origin would lead to an invariant specific carbon budget as a function of global warming”.

In retrospect that may have been too cryptic, so we have replaced this text with:

“We do not attempt to fit the slopes shown in Figure 2a, but instead seek an emergent relationship between cumulative emissions and global warming up to to the end of 2020, and the cumulative emissions at each future level of global warming”.

We have also added this text to the Methods section: “We look for emergent relationships between the cumulative emissions calculated from equation 2, and global warming. As we want to constrain the policy-relevant carbon budgets for the Paris climate targets, we work in terms of the specific carbon budget (i.e. the cumulative emissions per unit of global warming). This allows us to combine model differences and observational uncertainties in both global warming and cumulative emissions, into a single metric (see ‘Observational Constraint’ below)”.

3-3 It’s not real clear what your point is here. Specifically what is meant by “invariant”. If the linearity argument were strictly true, then given no change in emissions one would have no change in temperatures—so no offset. That is not the case, which is not a problem per se if one is only concerned with the increase. However, the slopes are not the same depending on the scenarios because they cover increasingly larger ranges in temperatures. That’s pretty clear from the NorESM models. As seen in Figure 2b, the slopes (PgC/C) are pretty different depending on the scenario, which is more of a statement that problem is non-linear, not linear since for higher scenarios, more carbon is getting added in.

Response: *This text has been rewritten in response to your other comments and no longer contains “invariant” (see response to 3-2).*

3-4 I don’t think it is appropriate to put multiple scenarios together. As can be seen from Fig. 2a, one would get different slopes (PgC/C) for each ESM ensemble for a given scenario. Consequently, the emergent constraint relative to a 2C change would be a function of scenario. Looking at Fig 2b, more closely if the ensembles were regressed on scenario, which is the right thing to do, then the slopes (and r^2) would be quite different. You need to justify why the scenarios should be thrown together (since they are correlated with each other). Why should we be looking at SSP585, which goes all the way pass 4C, when you’re only interested in 2C? If that slope is different (among ESMs) from the same calculation at SSP370, what is that telling us about the ESM TCRE? I’d suggest choosing one scenario (closest to 2C) and putting the other scenarios in supplemental. The fact that regardless of scenario, 300-700 PgC/C can’t all be correct—and observations can help through EC, is the main point from my perspective.

Response: *Carbon budgets as a function of global warming are especially useful as that they don’t require us to know which socioeconomic pathway we will be following in the future. This means that we can provide useful information with regard to emissions cuts, to say avoid exceeding 2°C of global warming, without having to know about other uncertain aspects of future socioeconomics. If instead we need to know which SSP we will be following before we can estimate the carbon budget, the utility of carbon budgets to guide policy is hugely diminished. In fact, the scenario sensitivity of our emergent constraints is weak, as we now show by calculating emergent constraints for each SSP separately. We include a new table in the supplementary material which summarizes those results, which we refer to from the main text: “Figure 2b includes all of our SSP scenarios as we aim to find*

an emergent constraint on the future carbon budget, independent of the unknown future SSP scenarios. However, our emergent constraints are actually rather insensitive to scenario, as we can show by calculating emergent constraints for each SSP separately (see Table 2)”.

We also want to check whether this linearity extends beyond the Paris targets, despite tipping points and non-linearities in ESMs – a linearity that is often debated at higher levels of global warming. This is why we extend our analysis out to 4°C of global warming. Some minor edits have been made to the text to make this rationale clearer.

3-5 Typo.

Response: “Ref⁸” corrected to “Ref 8”

3-6 I would like to see another vertical bar—say light green—of the prior model mean and standard deviation. It’s important to know what the relationship is between the model’s short-term response (PgC/C to 2020) and the model’s longer-term response (PgC at 2C). It’s combination of that short and long-term response (characterized by the r^2) and the observational uncertainty that lead to the emergent constraint.

Response: *The prior model mean and standard deviation are shown in Figure 3. We prefer not to repeat that information on Figure 2b which already contains a lot of information.*

3-7 That would be true if the emergent constraints were calculated for each scenario separately, but not together as in Fig. 2.

Response: *see response to 3-4.*

3-8 Typical for any kind of ensemble-based approach, the dependency of this estimate on the ensemble distribution itself is important. How robust are the solutions to the ensemble? This needs to be tested and included (could be referenced in supplemental). One easy way of doing that is to do a boot-strap approach (reduce the ensemble by 1 and then estimate the EC by adding/ removing 1 member at a time).

Response: *we have used all of the available ESM simulations that included all of the necessary outputs for all four SSP scenarios. The tightness of the emergent relationship shown in Figure 2b (such high r^2 values are very rare in emergent constraint studies) suggest a robustness of the emergent constraints presented here. However, we have now included additional Supplementary Material (Table S2) to show how the emergent constraint varies if we exclude one model at a time, as suggested.*

3-9 This is a bit confusing. Best-fit to what? From Fig. 2? Please be more specific to the data/plots. From the previous figures, the are different slopes derived from different target temperatures.

Response: *We mean here the best-fit straight-line to the red dots shown in Figure 4a, for all points greater than 1.2°C of global warming. This is explained in the caption of Figure 4, but to clarify we have now also added this text to the main manuscript: “(red dashed line in Figure 4a)”.*

3-10 If I understand this correctly, above a threshold, global warming is linear with the carbon accumulation per degree C. If it was perfectly linear, then Fig.4b would be a flat line. If that is correct, please say so.

Response: *To clarify we have added: “The specific carbon budget would be a flat line against global warming if global warming was strictly proportional to cumulative emissions (i.e. linear and with zero intercept). The specific carbon budget is however higher for lower values of global warming because of the cooling effect of anthropogenic aerosols in the second half of the 20th century”.*

4-1 As a matter of craft, the conclusions is pretty weak and doesn't really attempt to engage with how this work fits in the broader context. It doesn't really acknowledge any weaknesses, or how the approach could be improved in the future, or any recommendations/implications for policy/science. Needs to be beefed up.

Response: *'Conclusions' has been rewritten.*

7-1 I don't see the justification for calculating a temperature anomaly over just the last decade. The argument is based upon the historic temperature anomaly relative to historic cumulative emissions, not just the last decade. Need to see how sensitive the temperature anomaly calculation is to the time range.

Response: *All analysis is carried-out with respect to decadal mean temperature anomalies (to minimise the impacts of interannual variability). We are therefore just treating observations and models consistently. However, we must apologise for a typo in 'Methods, b) Data Processing' where the original manuscript incorrectly stated "we calculate centred 5 year means of $\Delta T(N)$ and $E(N)$ ". This has been corrected in the revised manuscript to "we calculate centred 10 year means of $\Delta T(N)$ and $E(N)$ ".*

7-2 Where does 0.12 C come from? To first order, I would expect it to be related to the distribution of temperature measurements. But, this needs to be clarified.

Response: *We have added text and a reference to clarify this point: "The figure of 0.12°C comes from IPCC AR6 WG1, Chapter 2 (Cross Chapter Box 2.3, Table 1, Footnote b) that states a 'likely uncertainty range of +/- 0.12°C' for the decadal mean global warming relative to the 1850-1900".*

7-3 I dont see how that is getting calculated. You can't just cite the whole IPCC. Provide a figure reference at least. Is this cumulative fossil fuel emissions or net emissions?

Response: *To clarify, we have added: "This estimate of uncertainty in the historical cumulative fossil fuel emissions is based-on Chapter 5 of the IPCC AR6 WG1 report⁸ (see Table 5.8, footnote b)".*

7-4 I don't think this is correct. You're calculating the ratio

$$Z_{PgC/C} = (\text{CumEmis} + \text{noise1}) / (\text{Tanomaly} + \text{noise2})$$

The resulting pdf is not a Gaussian with standard deviation that is the sum of the quadrature of noise1 and noise2. In other words, the quotient of two Gaussians is not a Gaussian.

Take a look at Simon, M. (2002). Probability distribution involving Gaussian random variables.

Boston: Kluwer Acad. Publishers. You can estimate it using a Monte Carlo, but there does exist (I believe) an analytic solution for the first and second moments.

Response: *To be frank, there is also no good reason to assume that the noise in the cumulative emissions or in the decadal mean temperature anomaly are Gaussians. So to us, it would seem to be spuriously accurate to assume that they are Gaussian and to use that assumption to derive an 'exact' non-Gaussian distribution for their ratio. Instead we have included a caveat: "In the absence of detailed information on the distributions of the random errors in the cumulative emissions and the decadal mean temperature anomalies, we assume that the uncertainty in the observed specific carbon budget is Gaussian distributed".*

8-1 This is not quite correct. Per Fig. 2, y is the cumulative emissions at 2C and x is the cumulative emission per change in temperature *of the models*, not the observations. There needs to be another conditional distribution that accounts for the PgC/C model distribution *given* observations of that distribution. Right now, the P(y|x) doesn't account for the correlation between the x and y of the implicit joint Gaussian distribution. In other words, P(y|x) is the same whether you have $r^2=1$ or $r^2=0.1$. I appreciate that approach has been used in the past, but there have been a number of publications since then that explicitly account for r^2 and the observational uncertainty, e.g.,

Sansom et al, 2021, Bretherton and Caldwell, 2020, Bowman et al, 2018; Please engage with those approaches and provide results on the dependency of these results to r^2 and the obs. error.

Response: We may have been unclear about the nature of our method here, so we have clarified by adding the following text: **“The emergent constraint $P(y)$ is therefore affected by both the quality of the emergent relationship $P(y|x)$ and the uncertainty in the observational constraint $P(x)$ ”.**

Our method has previously been compared with more sophisticated but less transparent Bayesian approaches (e.g. Samson et al., 2021) and has been found to give very similar results when the emergent relationship is strong (Nijse et al., 2020), as it is here. For example the PDF on the specific carbon budget for 2°C of global warming that we derive from ordinary-least-squares is very similar to the PDF that we derive from a hierarchical Bayesian approach:

We therefore prefer to stick with the more transparently simple OLS approach, although we acknowledge the reviewer’s point by adding the following text:

“Our statistical approach follows many other previous emergent constraint studies in applying an ordinary-least-squares (OLS) fit between the predictand and the predictor variable. More sophisticated Bayesian approaches have been proposed and applied in other studies (Bowman et al., 2018; Bretherton and Caldwell, 2020; Samson et al., 2021). These approaches yield very similar constraints to OLS when the emergent relationships are strong and linear (Nijse et al., 2020), as they are here.”

8-2 There’s every reason to believe that they would be correlated. You can say that you have to assume this because you don’t have a better solution, but the assumption itself is questionable.

Response: this is a fair comment, and one that really got us thinking. The correlation ρ_{xy} between x and y is of course encoded in an emergent relationship across models. Therefore, to avoid the possibility of making inconsistent assumptions concerning the correlation between the historical carbon budget and the total carbon budget, we have instead looked for a direct emergent constraint relating remaining carbon budgets for each level of global warming and the specific carbon budget up to the current day. The results of this approach are summarised in the new Figure 5.

11-1 This figure needs to be reworked. I can’t see the lower SSPs, they are all on top of each other from 1-2C and the lower ones don’t extend past out to the higher temperature anomalies.

Response: The SSPs are all on top of each other because the lines are largely independent of scenario (which is a very useful thing, see response to 2-8). The lower scenarios don’t extend out to the higher temperatures because global warming doesn’t get that far for the lower scenarios.

11-2 How does the pink range change as a function of correlation and observational noise?

Response: As the emergent relationship has a high r^2 , the uncertainty in the emergent constraint (i.e. the width of the pink range) varies approximately proportionally to the assumed uncertainty in the observational constraint.

11-3 Need a similar plot (in supplemental) of the Gaussian prior and observations of the historic emissions per unit warming.

Response: *The Gaussian prior for the specific carbon budget of the model ensemble is shown by the black line in Figure 3a. The observations of the historic emissions per unit of global warming are shown by the black stars in Figure 2a. We don't feel that we need to repeat this information in a separate plot.*

Response to Reviewer 2

We thank the reviewer for their detailed comments on our original submission. It has taken us some time to answer these, but we believe that they stimulated significant improvement in our revised paper, as outlined below. Reviewer comments are shown in normal black text, and our responses are in dark blue italics.

The authors introduce emergent constraints on the carbon budget as a function of global warming. By establishing a linear, emergent relationship between the specific carbon budget at a given level of global warming (y), and the specific carbon budget to “now” (x), the authors use an observational constraint to predict the cumulative carbon budget, with its uncertainty quantified.

Levels of global warming between 1°C and 4°C are considered, from which the remaining carbon in the budget beyond “now” (2020) can be predicted.

Motivation for this research is to reduce uncertainty in the latest CMIP6 Earth Systems Models projections. The authors find an emergent constraint that reduces uncertainties, which has the potential to contribute to climate-change policy-making. However, we have concerns about the statistical methodology used. This brings into question the veracity of the prediction and uncertainty calculations, both of which may affect the final results and yield incorrect conclusions.

Response: *We suspect that the explanation of our method wasn't clear enough. Our emergent constraint is deliberately on the specific carbon budget (i.e. the ratio of cumulative emissions to global warming) as this allows us to include uncertainty in both global warming and cumulative emissions. Our responses below, and the associated modifications to the manuscript, are designed to clarify our method.*

General comments

Statistical modeling of the emergent relationship and constraint is central to the manuscript. The authors make clear that there is uncertainty not only in y but also in x , which they specify through $P(x)$. As a consequence, the slope estimates obtained from Figure 2a are biased. There are papers that deal with this, the original one due to Bowman et al. (2018) who present a solution using a hierarchical statistical modeling framework. Hierarchical emergent constraint (HEC) modeling accounts for error in the x -variable and yields accurate and precise estimates, unlike those in Figure 2b. This framework has been applied in a number of articles; see, for example, Thackeray and Hall (2019), Barkhordarian et al. (2021), Shiogama et al. (2022), and Chen et al. (2022). The submitted manuscript needs to fit the emergent relationship based on HEC because the x -variable comes with uncertainty. We give more specific comments, by section, below.

Response: *We do not assume a proportionality between cumulative emissions and global warming, but instead look for an emergent relationship between the specific carbon budget to 2020 and the specific carbon budget at different levels of global warming. We are therefore not attempting to fit the slopes in Figure 2a as suggested by the reviewer's comment here. We have added this new text to make this clearer: “**We do not attempt to fit the slopes shown in Figure 2a, but instead seek an emergent relationship between cumulative emissions and global warming up to to the end of 2020, and the cumulative emissions at each future level of global warming**”. We have also added this text to the Methods section: “**We look for emergent relationships between the cumulative emissions calculated from equation 2, and global warming. As we want to constrain the policy-relevant carbon budgets for the Paris climate targets, we work in terms of the specific carbon budget (i.e. the cumulative emissions per unit of global warming). This allows us to combine model differences and observational uncertainties in both global warming and cumulative emissions, into a single metric (see ‘Observational Constraint’ below)**”.*

Introduction

- While the introduction is about remaining carbon budgets, most of the rest of the manuscript concentrates on the cumulative carbon budget (see conclusions and Table 1). The discussion around Figure 4c on lines 161-166 shows what remains as a function of global warming, and we suggest those results also be included in Table 1. The 30 PgC difference from IPCC6 (lines 179-180) is not significant. An emergent-constraint analysis needs to recognize properly the uncertainty in both y and x in order to provide policy-makers with accurate (unbiased) and precise (smallest uncertainty) results.

Response: *As suggested, we have added our new estimates of the remaining carbon budget to Table 1.*

Method

- The proposed method consists of several components: the modeled carbon budgets, a linear, emergent relationship, and the observational constraint. While discussion for the first one is well presented, those for the other two are quite limited. The linear emergent relationship deserves at least an equation, and it should be shown clearly how it is used to obtain the emergent constraint along with its uncertainty.

Response: *We do not assume a linear relationship between cumulative emissions and global warming (see response to ‘General Comments’ above), but we do find an approximately linear relationship between the specific carbon budget to 2020 and the carbon budget at various levels of global warming (as shown in Figure 2b). The method used to calculate the emergent constraint based-on this emergent relationship (and how well or otherwise it fits the data points) and the uncertainty in the observation of the x -variable, is as used in many previous studies (including the IPCC AR6 WG1 report, Chapter 5, Figure)and is described in ‘Methods d’, which we have extended to make the method clearer.*

- Please provide a clear definition of the observational constraint. Throughout the paper, it is either presented as a central estimate +/- errors, or a probability density function of the predictor in the Appendix. Sometimes the error bounds are not symmetric (e.g., Figure 4c); however, the +/- results given in the text assume they are symmetric.

Response: *We have now changed to quoting 66% confidence limits in square brackets after mean values, when the ranges are not strictly symmetric.*

Results

- In Bayesian statistics, the prior and posterior are usually distributions on the same variable, the former incorporating prior information and the latter being an update given new information. In fact, the statistical model in this manuscript is on the bivariate quantities (x, y) , where $P(x, y) = P(y | x)P(x)$, and interest is in the marginal distribution, $P(y) = \text{Integral}\{P(x, y)dx\}$. This is *not* an analogue of Bayes theorem (line 356). The theorem involves the inverse probability $P(x | y)$, which is not of great interest here for policy-making.

Response: *There is a prior distribution for the specific carbon budget which is derived from equal weighting of the models in the ensemble. The posterior distribution comes from the emergent constraint. Point taken though about “an analogue of Bayes theorem”, so we have removed that in the revised manuscript.*

- The authors use “prior” for $P(x)$ in Figure 3, which will cause confusion for readers.

Response: *We are not sure which text the reviewer is referring to here as we can’t find any instances of “prior” in the paper. However, the distributions shown in Figure 3 are prior (black) and posterior distributions (red) of $P(y)$, rather than $P(x)$.*

- Why did the authors use median instead of mean for the results based on emergent constraints?The uncertainty calculations for medians is different than that for means;

they cannot be used interchangeably, since the sample median has a different standard error than that of the sample mean.

Response: *as our emergent constraints give approximately symmetric uncertainties, the means and median values are very similar. However, it is true that our uncertainty estimates applies to the mean so we have made the change from 'median' to 'mean' in the revised manuscript.*

Conclusions

- The remaining carbon budget will be “spent” over time. In the 30 years between 2020 and 2050 (for example), what is the target spend-rate for 1.5°C, 2.0°C, and 2.5°C? How does that compare to the spend-rates up to 2020, where uncertainties should be incorporated into the comparison? These might be questions to address using the emergent-constraint analysis.

Response: *as suggested we have added text to the conclusions that compares these 'spend rates'.*

Appendix

- Uncertainty calculation for specific carbon budget (line 346). Using an additive formula to calculate uncertainty is suitable when one wants to compute $\text{var}(U + V)$. However, it is *not* true that $\text{var}(U + V)/[E(U + V)]^2 = \text{var}(U)/[E(U)]^2 + \text{var}(V)/[E(V)]^2$. This is in effect what the authors are proposing on line 346; one has to be careful when combining relative uncertainties.

Response: *here we are calculating an uncertainty in U/V not U+V, so combining the fractional uncertainty in U and V in quadrature is correct (see for example:*

https://en.wikipedia.org/wiki/Propagation_of_uncertainty#cite_note-17).

- The formulas for the uncertainty $\sigma_f(x)$ on lines 365–374 assume that x is known with certainty. However, x has uncertainty, and the authors estimate the uncertainty on y taking into account the uncertainty on x , as specified on lines 360–363. What then is the relationship between what is presented on lines 360–363 and lines 365–374? The two derivations do *not* result in the same uncertainty measure. This is another example of the need to use HEC from the beginning, where the emergent-relationship model has uncertainty in x , and every formula takes that into account.

Response: *The approach used here to derive emergent constraints has been very widely used and is described elsewhere (see for example Cox et al., 2013; Kwiatkowski et al., 2017; Cox et al., 2018; Williamson et al., 2021; IPCC AR6 WG1 Chapter5). The equation shown on line 365 is $P\{y|x\}$ which means the probability of y given x . However, we do not assume that we know x exactly. Instead $P\{x\}$ comes from the observational constraint and its uncertainties. We have added the following sentence extended 'Methods d)' to make this clearer: “**The emergent constraint $P(y)$ is therefore affected by both the quality of the emergent relationship $P(y|x)$ and the uncertainty in the observational constraint $P(x)$ ”.***

- Uncertainty calculation for the remaining carbon budget. Notice that $remain = total - hist$ or $r = y - x$. The authors claim that r and x are independent, which is almost certainly not true; since the total y is constrained for a given level of global warming, r and x will be negatively correlated. The appropriate calculation is

$$\sigma_r^2 = \sigma_y^2 + \sigma_x^2 - 2\rho_{xy}\sigma_x\sigma_y$$

where σ_y^2 and σ_x^2 are the variances of y and x , and ρ_{xy} is the correlation between x and y . In the manuscript, ϵ_{total}^2 and ϵ_{hist}^2 are estimates of σ_y^2 and σ_x^2 , respectively. The authors still need to estimate the third term.

Response: *this is a fair comment, and one that really got us thinking. The correlation ρ_{xy} between x and y is of course encoded in an emergent relationship across models. Therefore, to avoid the possibility of making inconsistent assumptions concerning the correlation between the historical*

carbon budget and the total carbon budget, we have instead looked for a direct emergent constraint relating remaining carbon budgets for each level of global warming and the specific carbon budget up to the current day. The results of this approach are summarised in the new Figure 5. We believe that this additional emergent constraint adds significantly to our study and we thank the reviewer for the query that provoked this improvement.

Minor comments

- Line 100: “based-on” → based on
- Line 155: “best-fit” → ordinary-least-squares?
- Line 168: “K” → °C (and throughout the manuscript)
- Line 301: “Timeseries” → Time series
- Lines 319, 328: “annual mean, global mean” → annual mean, global total? (Note: The unit of flux is “PgC/time/area”. To obtain the unit of “PgC/time” that the authors use, a total over “area” is needed. This applies to global, land, ocean, or any region of interest).

Response: *all these changes have been made in the revised manuscript, as suggested.*

REVIEWER COMMENTS

Reviewer #3 (Remarks to the Author):

The authors employ the linearity of TCRE (Transient Carbon Response to Cumulative Carbon Emissions) to constrain the remaining carbon budget for various warming levels. Their analysis is straightforward, utilizing observational temperature records and cumulative carbon emissions data from the Global Carbon Project to derive specific carbon budgets per 1°C of warming. This approach is applied to both historical runs of CMIP6 and various future scenarios. Due to the linearity of TCRE in CMIP6 Earth System Models (ESMs), the past can effectively inform the future in the model world. The authors conclude that their Emergent Constraint (EC) estimate for carbon budgets at specific warming levels exceeds the multi-model mean.

In essence, this study primarily involves a comparison of carbon budgets based on inverted TCRE for different warming levels. Given the linearity of TCRE in nearly all CMIP6 models (e.g., Williams et al., 2020, Environmental Research Letter), the Emergent relationship between specific past carbon budgets and future budgets must also exhibit linearity within the CMIP6 ensemble (e.g., Winkler et al., 2019, Earth System Dynamics). Therefore, one could streamline the analysis by linearly extrapolating the observed relationship (black line as depicted in Figure 2a, e.g., using a linear fit and disregarding values for the first 250 PgC due to natural variability). This extrapolation would closely follow the MPI-ESM estimate (slightly lower) and produce the same constraint (as shown in Figure 2b). As linearity prevails, it's expected that the results based on this simple extrapolation and the Emergent Constraint are more or less the same. In other words, this would represent a similar analysis, as the authors rely on the TCRE linearity, which is well-established for the latest ESM versions.

Therefore, this manuscript appears to provide limited additional insights to the field. Figure 1 and Figure 2a primarily plot model outputs from CMIP6, which can be found in numerous other papers. Moreover, the presented Emergent Constraint essentially involves a linear extrapolation of the observed relationship between warming and cumulative emissions. This constraint doesn't significantly deviate from, or extend well beyond the uncertainty bounds established by the IPCC. Consequently, while the manuscript is suitable for a more specialized journal, it might not align with the standards of Nature Communications.

Additional Comments:

1. Since the analysis is relatively straightforward, it would be helpful to understand why CMIP5 models were not included in the study.
2. Considering that SSPs comprise various forcing agents beyond CO₂, which could contribute to uncertainty, it's worth noting that this might further increase the overall uncertainty and potentially weaken the constraint.
3. In lines 21-23, it would be beneficial to clarify how the EC values were deemed significantly larger, given that the mean values fall within the uncertainty of the emergent constraint. Which test was used to test for significance?
4. In lines 27-29, please note that the observation regarding TCRE in CMIP6 is not a novel finding, as it has been addressed in prior studies, such as Williams et al. 2020 (Environmental Research Letters).
5. In line 41, it's advisable to remove the word "Unfortunately" as it may sound judgmental and does not

contribute to the statement. Simply stating, "The carbon budgets consistent with the Paris Targets remain uncertain," suffices.

6. In line 94, specify whether the study used the esmHistorical run (coupled carbon cycle) or the historical run, and provide justification for the choice.

7. Constructing land-sea masks can be easily accomplished based on model output (e.g., using land-sea masks generated from land GPP data). Robust results based on Emergent Constraints benefit from the inclusion of more models, especially when compared to the model mean (as mentioned in the Abstract, "significantly higher than model mean").

8. In line 290, correct the typo to "previously."

9. In Figure 1, include a heading indicating that this pertains to the historical + ssp245 scenario.

Reviewer #4 (Remarks to the Author):

This study applies emergent constraint on the specific carbon budget by the ratio of cumulative emissions to global warming. The results show constrained cumulative carbon budgets for the Paris targets of 1.5°C and 2°C are larger than the CMIP6 model mean values, with uncertainty range reduced. Overall, I think this paper is well written and the results make a significant contribution to existing literature.

General comments

1. As in Table 2, some models with close versions involved in constructing the emergent constraint, such as CESM2 and CESM-WACCM, CanESM5 and CanESM5-CanOE, NorESM2-LM, and NorESM2-MM. This raises concerns about the independence of the models in your emergent constraint, as these similarities might potentially reinforce the emergent relationship with redundancy.

2. With only 9 models in use, there's a concern that the emergent relationship may be overconfident. If more models were included in the analysis, it raises the question of whether the constrained results would still be robust.

3. The 'leave one out' analysis (Table 3) may not be sufficient as this test can not rule out the risk of an overconfident emergent relationship. A more robust approach for systematically assessing emergent relationship is cross-validation test, as demonstrated by studies like Brunner et al. (2020) and Ribes et al. (2021) by comparing pseudo observations and constrained projections produced by emergent constraint method.

Minor comments

1. References 17 and 18 appear to be identical, and the same for References 19 and 30. Please check.

Reference:

Aurélien Ribes et al. ,Making climate projections conditional on historical observations. *Sci. Adv.*7,eabc0671(2021).DOI:10.1126/sciadv.abc0671

Brunner, L., Pendergrass, A. G., Lehner, F., Merrifield, A. L., Lorenz, R., & Knutti, R. (2020). Reduced global warming from CMIP6 projections when weighting models by performance and independence. *Earth System Dynamics*, 11(4), 995-1012.

Response to Reviewer 3

We thank the reviewer for their comments on our revised submission. Below, each reviewer comment is shown in normal black text, with our response immediately following in dark blue italics.

3.1 The authors employ the linearity of TCRE (Transient Carbon Response to Cumulative Carbon Emissions) to constrain the remaining carbon budget for various warming levels. Their analysis is straightforward, utilizing observational temperature records and cumulative carbon emissions data from the Global Carbon Project to derive specific carbon budgets per 1°C of warming. This approach is applied to both historical runs of CMIP6 and various future scenarios. Due to the linearity of TCRE in CMIP6 Earth System Models (ESMs), the past can effectively inform the future in the model world. The authors conclude that their Emergent Constraint (EC) estimate for carbon budgets at specific warming levels exceeds the multi-model mean.

***Response:** This is not quite what we have done. We do not a priori assume linearity between cumulative emissions and global warming, and we constrain the carbon budget for different levels of global warming including the uncertainties in non-CO₂ forcing factors. Our specific carbon budgets are therefore not the inverse of the CO₂-only TCRE as the reviewer suggests, but instead the inverse of the effective TCRE, which includes the effects of non-CO₂ forcing factors. We have added the following text to make this clearer:*

“We do not a priori assume a linear relationship between emissions and global warming. Instead we look for an emergent relationship across models between the cumulative carbon emissions per degree of global warming up to the current day, and the specific carbon budgets for different future levels of global warming. As we are interested in constraining the global carbon budgets for the Paris climate targets of 1.5°C and 2°C, we deliberately span scenarios with different future trajectories of non-CO₂ forcing factors. As a result, our specific carbon budgets are not the inverse of the CO₂-only ‘Transient Climate Response to Emissions’ (TCRE), but are instead the inverse of the effective TCRE, which includes the effects of non-CO₂ forcing factors. The latter is much more useful for UNFCCC climate policy, which needs to account for uncertainties in future atmospheric concentrations of non-CO₂ greenhouse gases and aerosols, as well as CO₂”.

3.2 In essence, this study primarily involves a comparison of carbon budgets based on inverted TCRE for different warming levels. Given the linearity of TCRE in nearly all CMIP6 models (e.g., Williams et al., 2020, Environmental Research Letter), the Emergent relationship between specific past carbon budgets and future budgets must also exhibit linearity within the CMIP6 ensemble (e.g., Winkler et al., 2019, Earth System Dynamics). Therefore, one could streamline the analysis by linearly extrapolating the observed relationship (black line as depicted in Figure 2a, e.g., using a linear fit and disregarding values for the first 250 PgC due to natural variability). This extrapolation would closely follow the MPI-ESM estimate (slightly lower) and produce the same constraint (as shown in Figure 2b). As linearity prevails, it's expected that the results based on this simple extrapolation and the Emergent Constraint are more or less the same. In other words, this would represent a similar analysis, as the authors rely on the TCRE linearity, which is well-established for the latest ESM versions.

***Response:** The simple extrapolation of the historical record has of course been done and was already referenced in our paper (Millar and Friedlingstein, 2018). This extrapolation requires assuming linearity between cumulative emissions and global warming, which is questionable given the other uncertain forcing factors that affect global warming (see Figure 4b which shows how the specific carbon budget decreased through the latter half of the 20th century as a result of a declining atmospheric aerosol loading). It is also difficult to separate out the impacts of climate variability on the effective TCRE, because there is essentially only one realisation of the historical record. In contrast, our approach naturally accounts for non-CO₂ forcing factors (by including multiple SSPs)*

and accounts for climate variability (by including multiple models), to produce estimates that are consistent with both the model ensemble and the observational record.

3.3 Therefore, this manuscript appears to provide limited additional insights to the field. Figure 1 and Figure 2a primarily plot model outputs from CMIP6, which can be found in numerous other papers. Moreover, the presented Emergent Constraint essentially involves a linear extrapolation of the observed relationship between warming and cumulative emissions. This constraint doesn't significantly deviate from, or extend well beyond the uncertainty bounds established by the IPCC. Consequently, while the manuscript is suitable for a more specialized journal, it might not align with the standards of Nature Communications.

Response: *We disagree for the reasons outlined above, and also because our study includes both models and observations in a transparent way. This is in contrast to more standard methods used to estimate carbon budgets (Canadell et al., 2021) which ignore certain aspects of ESMs (e.g. the ESM range of climate sensitivity to CO₂) and yet include others (e.g. the ESM range of carbon cycle feedback factors).*

3.4 Additional Comments:

3.4.1. Since the analysis is relatively straightforward, it would be helpful to understand why CMIP5 models were not included in the study.

Response: *We have now tested the robustness of our emergent constraint by repeating for the CMIP5 model ensemble as suggested by the reviewer. The following text explaining this has been added to the main manuscript: “We have focussed our study on the CMIP6 ESMs, because: (a) these are most recent generation of models that have undertaken the required model runs; (b) in CMIP6 many more models completed all of the relevant runs for ssp126, ssp245, ssp370, ssp585, which allows us to cleanly account for the effects of different future scenarios of non-CO₂ factors. Nevertheless, to test the robustness of our findings we repeated our emergent constraints using climate-carbon cycle runs from the previous CMIP5 generation of models. Figure S7 in the Supplementary Material shows the equivalent emergent constraint on the carbon budget for 2°C based on the available CMIP5 model runs. Although the emergent relationship for the CMIP5 models is weaker (in part because it is based on 8 data points rather than 36), it leads to a very similar best estimate for the carbon budgets (e.g. for 2°C of global warming: 1045 PgC from CMIP5; 1048 PgC from CMIP6)”.*

3.4.2. Considering that SSPs comprise various forcing agents beyond CO₂, which could contribute to uncertainty, it's worth noting that this might further increase the overall uncertainty and potentially weaken the constraint.

Response: *Our approach spans four very different SSP scenarios and therefore implicitly accounts for the uncertainties associated with non-CO₂ forcing factors (see our response above to 3.1).*

3.4.3. In lines 21-23, it would be beneficial to clarify how the EC values were deemed significantly larger, given that the mean values fall within the uncertainty of the emergent constraint. Which test was used to test for significance?

Response: *Apologies, this was a colloquial use of “significantly”, which we have removed from the revised manuscript.*

3.4.4. In lines 27-29, please note that the observation regarding TCRE in CMIP6 is not a novel finding, as it has been addressed in prior studies, such as Williams et al. 2020 (Environmental Research Letters).

Response: *Citations are not allowed in the abstract of a Nature Communications paper. However, the Williams et al. (2020) paper is already cited in the ‘Results’ section.*

3.4.5. In line 41, it's advisable to remove the word "Unfortunately" as it may sound judgmental and does not contribute to the statement. Simply stating, "The carbon budgets consistent with the Paris Targets remain uncertain," suffices.

Response: *We have replaced "Unfortunately" with "However".*

3.4.6. In line 94, specify whether the study used the esmHistorical run (coupled carbon cycle) or the historical run, and provide justification for the choice.

Response: *As requested, we have added: "To be consistent with the prescribed CO₂ runs that we use for the ssp scenarios, we use the CMIP6 historical runs (with prescribed atmospheric CO₂) rather than the CMIP6 esm-hist runs (which calculate the atmospheric CO₂ interactively based-on prescribed CO₂ emissions)."*

3.4.7. Constructing land-sea masks can be easily accomplished based on model output (e.g., using land-sea masks generated from land GPP data). Robust results based on Emergent Constraints benefit from the inclusion of more models, especially when compared to the model mean (as mentioned in the Abstract, "significantly higher than model mean").

Response: *Our choice of CMIP6 models was mainly to ensure that each model variant had simulated both the land and ocean carbon cycle in all of the ssp scenarios (ssp126, ssp245, ssp370, ssp585). This provided us with 36 datapoints (9 models x 4 scenarios) for our emergent constraints, and the tightest emergent relationships that our authorship team had ever seen (e.g. Figure 2b).*

3.4.8. In line 290, correct the typo to "previously."

Response: *done.*

3.4.9. In Figure 1, include a heading indicating that this pertains to the historical + ssp245 scenario.

Response: *for clarity, we have instead included this information in the figure caption.*

Response to Reviewer 4

We thank the reviewer for their comments on our revised submission. Below, each reviewer comment is shown in normal black text, with our response immediately following in dark blue italics.

4.1 This study applies emergent constraint on the specific carbon budget by the ratio of cumulative emissions to global warming. The results show constrained cumulative carbon budgets for the Paris targets of 1.5°C and 2°C are larger than the CMIP6 model mean values, with uncertainty range reduced. Overall, I think this paper is well written and the results make a significant contribution to existing literature.

***Response:** many thanks for this endorsement of our work!*

4.2 As in Table 2, some models with close versions involved in constructing the emergent constraint, such as CESM2 and CESM-WACCM, CanESM5 and CanESM5-CanOE, NorESM2-LM, and NorESM2-MM. This raises concerns about the independence of the models in your emergent constraint, as these similarities might potentially reinforce the emergent relationship with redundancy.

***Response:** Our choice of CMIP6 models was mainly to ensure that each model variant had simulated both the land and ocean carbon cycle in all of the ssp scenarios (ssp126, ssp245, ssp370, ssp585). This provided us with 36 datapoints (9 models x 4 scenarios) for our emergent constraints, and the tightest emergent relationships that our authorship team has ever seen (e.g. Figure 2b). As a result of the tightness of the emergent relationship, we can exclude repeated models and still get very similar results (see new Table 3 which now includes a row with CESM-WAGCM, CanESM5-CanOE, NorESM2-MM all excluded).*

4.3 With only 9 models in use, there's a concern that the emergent relationship may be overconfident. If more models were included in the analysis, it raises the question of whether the constrained results would still be robust.

***Response:** see response to 4.2 above. We have also added an equivalent emergent constraint on the carbon budget for 2°C based on the available CMIP5 model runs (see new Figure S7). Although the emergent relationship for the CMIP5 models is weaker (in part because it is based on 8 data points rather than 36), it leads to a very similar best estimate for the carbon budget.*

4.4 The 'leave one out' analysis (Table 3) may not be sufficient as this test can not rule out the risk of an overconfident emergent relationship. A more robust approach for systematically assessing emergent relationship is cross-validation test, as demonstrated by studies like Brunner et al. (2020) and Ribes et al. (2021) by comparing pseudo observations and constrained projections produced by emergent constraint method.

***Response:** thank you for the references. These approaches are beyond the scope of this study, but we have included an additional test of robustness based on CMIP5 models (see response to 4.3 above).*

4.5 Minor comments

1. References 17 and 18 appear to be identical, and the same for References 19 and 30. Please check.

***Response:** Apologies, these typos have been corrected in the revised manuscript.*